# Self-reported concussion history is not related to cortical volume in college athletes

Douglas H. Schultz[1,2]*, Heather C. Bouchard[1,2], Michelle C. Barbot[3], Julia M. Laing-Young[1,2], Amanda Chiao[4], Kate L. Higgins[5], Cary R. Savage[1,2], & Maital Neta[1,2]

1 Center for Brain, Biology and Behavior, University of Nebraska-Lincoln, Lincoln, Nebraska, United States of America, 2 Department of Psychology, University of Nebraska-Lincoln, Lincoln, Nebraska, United States of America, 3 University of Nebraska Medical Center, Omaha, Nebraska, United States of America, 4 Department of Surgery, Texas Tech University Health Sciences Center El Paso, El Paso, Texas, United States of America, 5 Department of Athletics, University of Nebraska-Lincoln, Lincoln, Nebraska, United States of America

* dhschultz@unl.edu

**Data availability statement:** Participants in this study did not consent to the release of their data

## Abstract

The long-term consequences of concussion are still being uncovered but have been linked to disruptions in cognition and psychological well-being. Previous studies focusing on the association between concussion history and structural changes in the brain have reported inconsistent results. We sought to examine the effect of concussion history on cortical volume with a focus on functional networks. These networks are associated with many of the functions that can be disrupted in those with an extensive concussion history. We collected baseline behavioral data including the Immediate Post-Concussion Assessment and Cognitive Testing, a self-report measure of the number of diagnosed concussions, and structural MRI in college athletes (n=296; 263 men and 33 women, age range 17-24). Behavioral measures were collected by members of the Department of Athletics concussion management team using a standardized protocol as part of their on-boarding process. Collegiate athletes in the present study who self-reported concussion history did not report different baseline symptoms and did not exhibit consistent differences in cognitive performance relative to those who reported no concussion history. We found that concussion history was not related to cortical volume at the network or region level, even when we compared participants with two or more concussions to those with no concussion history. We did identify relationships between cortical volume in the visual network and dorsal attention network with cognitive performance. In addition to comparing cortical volume between individuals with and without reported concussion history, we also examined whether cortical volume changes could be observed within individuals from baseline to acutely following concussion. We found that network level cortical volume did not change within subjects from baseline measurement to acutely post-concussion. Together, these results suggest that both self-reported concussion history and acute concussion effects are not associated with changes in cortical volume in young adult athletes.

to a third-party repository. Therefore, we are unable to publicly archive data due to the conditions of our ethics approval. Readers inquiring about the current data should contact the corresponding author (Douglas H. Schultz) or Aron Barbey (abarbey2@unl.edu), the Director of the Center for Brain, Biology and Behavior, who is also responsible for project oversight. Access can only be granted in accordance with ethical procedures governing the reuse of sensitive data.

**Funding:** Funding for this study was provided by the Great Plains IdeA-CTR (1U54GM115458) (awarded to MN) and the UNL Office for Research and Economic Development (awarded to CS). The funders had no role in study design, data collection and analysis, decision to publish, or preparation of the manuscript.

**Competing interests:** The authors have declared that no competing interests exist.

## Introduction

Each year, tens of thousands of adolescents and young adults experience a sport-related concussion, a form of mild traumatic brain injury [1]. Concussion can result in cognitive, somatic, affective, and sleep symptoms. In most cases, clinical recovery from concussion symptoms occurs within 14 days [2]. Distinct from recovery from acute concussion, concussion history refers to how many concussions a person has experienced in their life. People with a self-reported history of concussion may be at elevated risk for future injury [3,4], and prolonged recovery [5,6]. Concussion history has also been associated with other adverse outcomes related to cognition and motor function [7], increased rates of depression [8], and increased risk of stroke [9]. The possible long-term effects of concussion are of increasing interest and an open area of inquiry [10].

In conjunction with clinically relevant changes related to concussion history, research suggests that concussion history is associated with alterations in white matter microstructure [11–13], functional connectivity [11,14–16], and electrophysiological measures [17–19]. While concussion history appears to be related to these measures of brain structure and function, it is currently unclear whether concussion history results in morphologic changes in gray matter. Gray matter morphologic measures change throughout development [20,21] and in neurodegenerative disorders like chronic traumatic encephalopathy [22], Alzheimer's disease [23], and Parkinson's disease [24]. If cortical morphology is influenced by neurodegenerative disorders, the possible link between concussion history and these types of disorders later in life [25] may indicate that concussion history results in morphological changes in the brain and could serve as an objective, quantitative marker for cumulative concussion history.

Previous studies have examined the effect of concussion history on brain morphology; however, the literature on this topic is inconsistent. Some of these inconsistencies may be related to the heterogeneous nature of concussion, focusing on concussion history specifically in athletes, in military personnel, in the general population, and possible differences in the mechanism of injury. Early studies suggested that cortical thickness or volume is generally decreased in those with concussion history across the whole brain [26], and frequently within the frontal cortex [27–31]. Other studies have identified increases in cortical thickness or volume with longer intervals between injury and assessment [32,33], and at more acute time points following injury [34,35]. Recent studies with larger sample sizes have shown that concussion history is not associated with any significant differences in cortical thickness or volume [36–40] raising questions about the reliability of the earlier work with smaller sample sizes.

Many of the adverse outcomes related to concussion history can be viewed through the lens of network neuroscience. In fact, concussion is increasingly being considered as an injury characterized by disruptions to structural and functional brain connectivity [41–45]. These network disruptions have been linked with many of the behavioral outcomes of concussion history including cognitive impairment [46–48], emotion disruptions including depression [49–51], and changes in motor function [52]. Most previous studies have examined brain morphology in the context of concussion history at the vertex level or in regions of interest. The goal of the current project was to determine the effects of self-reported concussion history on brain morphology, specifically examining how network-level changes in cortical volume were related to concussion history in a large sample of collegiate athletes. This project will also adjudicate on the mixed findings in the literature relating concussion history and brain morphological outcomes.

## Materials and methods

### Participants

Two hundred ninety-six (men's football n = 263; women's soccer n = 33) players from the University of Nebraska-Lincoln (UNL) participated in the study (mean age = 19.46 years old, SD = 1.65, range = 17-24). The UNL IRB approved the study. Written consent was obtained from participants. Data was collected from June 2018 through May 2022. Clinical baseline assessments consisting of cognitive performance with the Immediate Post-Concussion Assessment and Cognitive Testing (ImPACT), which includes a baseline symptom scale [53], five cognitive composite scores based on task performance (verbal memory, visual memory, visual motor speed, reaction time, and impulse control), along with a self-report measure of the number of diagnosed concussions experienced prior to joining UNL Athletics (Fig 1). These measures were collected by members of the Department of Athletics concussion management team using a standardized protocol. Participants were divided into two groups based on their self-reported history of concussion. One group self-reported zero concussions prior to their baseline assessment at UNL (n=194), and the second group self-reported at least one concussion prior to their baseline assessment at UNL (n=102). The first year of the study (football, 2018; soccer, 2019) all student-athletes underwent a baseline MRI scan. Since that time, incoming student-athletes underwent a baseline scan prior to participating in their sport. A subset of participants (n = 44, 40 males, 4 females) received additional scans if they were diagnosed with a sports-related concussion while competing at UNL. One set of scans was collected within approximately 48 hours of a diagnosed concussion and a second was collected during their return to play protocol immediately prior to returning to contact.

### MRI acquisition

MRI data were collected on a 3T Siemens Skyra scanner located within the Center for Brain Biology and Behavior at UNL. As part of a larger project, T1-weighted (TR = 2530 ms, flip angle = 7, FOV = 256 mm/100% phase, 176 slices, slice thickness = 1 mm), resting-state fMRI, and diffusion scans were collected. For this report, we focus on the T1-weighted data. MRI acquisition collected post-concussion and during return to play were identical to those collected at baseline.

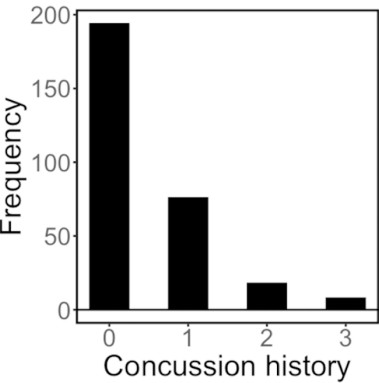

**Fig 1. Historgram of self-reported concussion history upon baseline assessment.**

## MRI preprocessing

The FreeSurfer toolbox (Version 5.3; https://surfer.nmr.mgh.harvard.edu/) was used with the default settings for "recon-all" to process anatomical brain images [54–56]. Structural image processing consisted of skull stripping, motion correction, affine transformation, correction for inhomogeneities, and parcellation of the cerebral cortex into gyral and sulcal regions [57], (Fig 2A). For region-level analyses, we calculated the mean cortical volume for each of the 74 regions in the Destrieaux atlas across hemispheres for each participant. Cortical volume for each of the 74 regions was normalized by dividing the volume measure for each region by the estimated total intracranial volume, which is an index of the participants' intracranial volume in native space as a function of the atlas scaling factor [58]. Region-level statistical tests were conducted in Matlab [59] and JASP [60].

## MRI network analysis

The cortex was parcellated into seven different networks (visual, somatomotor, dorsal attention, ventral attention, limbic, control, and default mode) [61] which were derived from a stable clustering algorithm applied to resting-state fMRI data from 1000 healthy control participants. The Yeo atlas was transformed to individual participant space using the mri_surf2surf program in FreeSurfer, (Fig 2B). Then cortical volume statistics for each network were calculated for each hemisphere for each participant using the mri_anatomical_stats program in FreeSurfer. The mean cortical volume measure across hemispheres was calculated for each participant. Finally, cortical volume measures were normalized by dividing the volume measure for each network by the estimated total intracranial volume. Network-level statistical tests were conducted in Matlab [59] and JASP [60].

## Results

### No differences on symptom report or cognitive measures as a function of concussion history

First, we examined whether there were differences in baseline symptoms or cognitive performance (as measured by ImPACT), between participants who reported a previous concussion and those who reported no previous concussion. The data on symptoms was not normally distributed so we used the nonparametric Mann-Whitney U-test and false discovery rate (FDR) [64] to correct for multiple comparisons across the ten measures. We observed a difference

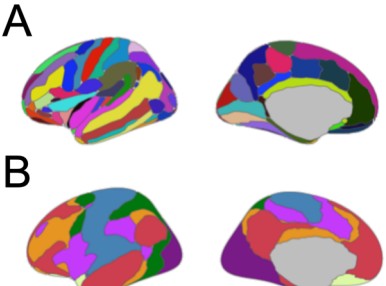

**Fig 2. The atlases used for cortical volume comparisons.** A. The Destrieux atlas [57] was used for regional comparisons of cortical volume. B. The 7 Network Yeo atlas was used for network-level comparisons of cortical volume [61]. Visualizations were created in R Studio [62] using the ggseg package [63].

between groups on the reaction time cognitive composite from the ImPACT, p = 0.04. We did not observe differences between groups on total symptoms, any of the self-reported symptom clusters (cognitive, somatic, affective, and sleep) [65], or any other cognitive composite scores from the ImPACT (verbal memory, visual memory, visual-motor speed, and impulse control), FDR-corrected ps > 0.09 (Fig 3). In addition to null hypothesis significance testing, we also used a Bayesian Mann-Whitney U-test to examine the degree of evidence for the alternative and null hypothesis. The Bayesian statistics showed anecdotal evidence for a group difference on the visual memory, BF10 = 2.664, and reaction time, BF10 = 1.081 cognitive composites. There was anecdotal evidence for the null hypothesis on visual-motor speed, BF10 = 0.451, somatic symptoms, BF10 = 0.339, and affective symptoms, BF10 = 0.346. There was moderate evidence for the null hypothesis on verbal memory, BF10 = 0.326, impulse control, BF10 = 0.136, total symptoms, BF10 = 0.142, cognitive symptoms, BF10 = 0.173, and sleep symptoms, BF10 = 0.137. Despite differences in concussion history, we did not observe consistent evidence from the null hypothesis significance testing and Bayesian statistics approaches (with a moderate level of evidence cut-off) for any differences in self-reported symptoms or cognitive performance measures.

## No differences in network cortical volume related to concussion history

We compared mean cortical volume in each of the seven networks between participants who reported concussion history and those who reported no history of concussion using a Mann-Whitney U-test. We did not observe any differences in cortical volume even with uncorrected p-values, smallest p = 0.067. In addition to null hypothesis significance testing, we also used a Bayesian Mann-Whitney U-test to examine the degree of evidence for the null hypothesis (that cortical volume was not different between those with and without self-reported concussion history). We found moderate evidence, BF10 values between 0.155 and 0.136 [66], for the null hypothesis in six networks (visual, somatomotor, dorsal attention, ventral attention, control, and default mode, Table 1). We found anecdotal evidence for the null hypothesis in the limbic network, BF10 = 0.483. Finally, we used a Spearman correlation to examine the relationship between cortical volume in each network and the number of concussions reported by each participant. There was not a significant relationship between concussion history and cortical volume in any of the seven networks, smallest uncorrected p = 0.075.

## No difference in regional cortical volume related to concussion history

Although we did not find any differences in cortical volume at the network-level, it is possible that there are volume changes at a smaller spatial scale that are lost when looking at whole networks. Therefore, we ran the same comparisons for cortical volume in a larger number of spatially smaller, and anatomically defined regions from the Destrieux atlas. We compared the mean cortical volume in each of the 74 regions between participants who reported concussion history and those who reported no history of concussion using a Mann-Whitney U-test. There were no significant differences in cortical volume between groups after FDR correction (Table 2).

## Concussion history of two or more is not associated with differences in cortical volume

We did not observe any differences in cortical volume when comparing participants reporting any history of concussion (one or more concussions) to participants reporting no history of concussion. However, it is possible that we may have identified differences if we compared

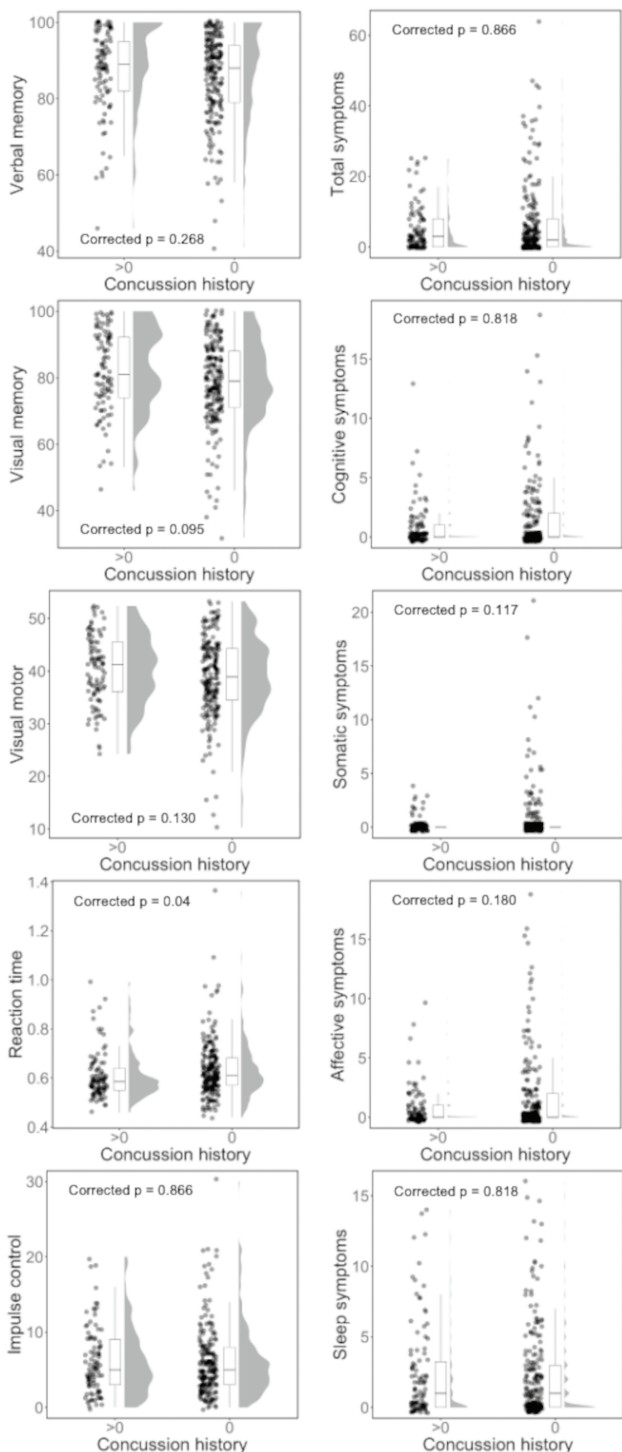

**Fig 3. Participants who report concussion history do not endorse more symptoms or show differences in cognitive performance (with the possible exception of reaction time) from those who report no concussion history.** Raincloud plots for five measures related to symptoms (total symptoms, cognitive, somatic, affective, and sleep symptoms) in the right panels, and five composite scores for the ImPACT (verbal memory, visual memory, visual motor speed, reaction time, and impulse control) in the left panels. For each raincloud plot the data for each participant is displayed on the left with a jitter. The middle portion is the boxplot for the data. The shape of the distribution of the data is plotted on the right. The FDR-corrected null hypothesis significance testing statistics are reported in each panel.

**Table 1. Comparison of cortical volume in networks for participants reporting concussion history with those reporting no concussion history.**

| Network | Mann-Whitney | | BF10 |
|---|---|---|---|
| | W | Uncorrected p | |
| Visual | 9720 | 0.862 | 0.133* |
| Somatomotor | 9359 | 0.727 | 0.136* |
| Dorsal Attention | 10323 | 0.295 | 0.201* |
| Ventral Attention | 9960 | 0.602 | 0.160* |
| Limbic | 8731 | 0.208 | 0.243* |
| Control | 10303 | 0.308 | 0.191* |
| Default Mode | 9209 | 0.571 | 0.132* |

*Moderate evidence for $H_0$.

the no concussion history group to participants who report a greater number of prior concussions. We compared cortical volume for participants who reported having two or more concussions before completing their baseline MRI scan at UNL using a Mann-Whitney U-test. There were 26 participants who reported more than one concussion in their lifetime (2+ concussions), and 200 reporting no history of concussion. There was no difference in cortical volume in any of the seven networks for the participants with two or more concussions relative to the no concussion history group, smallest p = 0.136.

## Cortical volume is related to cognitive measures

Although we did not identify any relationship between cortical volume and concussion history, we did find relationships between cortical volume and cognitive measures. Specifically, we examined the five cognitive composites from the ImPACT test (verbal memory, visual memory, visual motor speed, reaction time, and impulse control) using Spearman correlations because these were not all normally distributed. We used FDR to correct for multiple comparisons of cortical volume over the seven networks and five cognitive performance scores. We found that cortical volume in the visual network was positively correlated with verbal memory (rho = 0.1718, FDR-corrected p = 0.0265), visual memory (rho = 0.2364, FDR-corrected p = 0.0014), and visual motor speed (rho = 0.1662, FDR-corrected p = 0.0291), and negatively correlated with reaction time (rho = -0.1876, FDR-corrected p = 0.0207). Further, cortical volume in the dorsal attention network was positively correlated with visual memory (rho = 0.1792, FDR-corrected p = 0.023) (Fig 4).

## No differences in cortical volume related to acute concussion

Next, we compared effects of self-reported concussion history to acute effects following a concussion. Indeed, although we did not observe a relationship between concussion history and cortical volume, it is possible that changes in cortical volume related to concussion are evident closer to the time of injury and that they had largely returned to normal when we collected baseline measures in our sample. To address this possibility, we examined cortical volume in a subset of our sample who were diagnosed with a concussion during their time playing sports at UNL. These 44 participants received baseline scans when they entered the program, within approximately 48 hours of being diagnosed with a concussion (i.e., acute period), and during their return to play protocol immediately prior to returning to contact. We did not identify any changes in cortical volume in any of the seven networks from baseline to the acute

**Table 2. Comparison of cortical volume in Destrieux atlas regions for participants reporting concussion history with those reporting no concussion history.**

| Region label | Mann-Whitney | | |
| --- | --- | --- | --- |
| | W | Uncorrected p | FDR-corrected p |
| G and S frontomargin | 9549 | 0.6225 | 0.8871 |
| G and S occipital inf | 9877 | 0.9812 | 0.9903 |
| G and S paracentral | 10293 | 0.5691 | 0.8871 |
| G and S subcentral | 9018 | 0.2109 | 0.6503 |
| G and S transv frontopol | 9693 | 0.7745 | 0.9097 |
| G and S cingul-Ant | 10370 | 0.4968 | 0.8871 |
| G and S cingul-Mid-Ant | 10788 | 0.2017 | 0.6503 |
| G and S cingul-Mid-Post | 10796 | 0.1977 | 0.6503 |
| G cingul-Post-dorsal | 10811 | 0.1903 | 0.6503 |
| G and S cingul-Post-ventral | 11965 | 0.0031 | 0.2287 |
| G cuneus | 10896 | 0.1524 | 0.6503 |
| G front inf-Opercular | 10965 | 0.1261 | 0.6503 |
| G front inf-Orbital | 9345 | 0.4332 | 0.8871 |
| G front inf-Triangul | 10016 | 0.8622 | 0.9522 |
| G front middle | 10980 | 0.1209 | 0.6503 |
| G front sup | 9217 | 0.3337 | 0.7717 |
| G front Ins lg and S cent ins | 9690 | 0.7712 | 0.9097 |
| G insular short | 10581 | 0.3266 | 0.7717 |
| G occipital middle | 10323 | 0.5403 | 0.8871 |
| G occipital sup | 9605 | 0.6802 | 0.8871 |
| G oc-temp lat-fusifor | 9177 | 0.3059 | 0.7717 |
| G oc-temp med-Lingual | 9494 | 0.5681 | 0.8871 |
| G oc-temp med-Parahip | 8251 | 0.0189 | 0.3582 |
| G orbital | 8812 | 0.1223 | 0.6503 |
| G parietal inf-Angular | 9426 | 0.5041 | 0.8871 |
| G parietal inf-Supramar | 11156 | 0.0715 | 0.6503 |
| G parietal sup | 11158 | 0.0710 | 0.6503 |
| G postcentral | 10076 | 0.7954 | 0.9196 |
| G precentral | 10127 | 0.7397 | 0.9097 |
| G precuneus | 10244 | 0.6175 | 0.8871 |
| G rectus | 9869 | 0.9712 | 0.9903 |
| G subcallosal | 9636 | 0.7129 | 0.8942 |
| G temp sup-G T transv | 9196 | 0.3189 | 0.7717 |
| G temp sup-Lateral | 9685 | 0.7658 | 0.9097 |
| G temp sup-Plan polar | 9482 | 0.5565 | 0.8871 |
| G temp sup-Plan tempo | 9432 | 0.5096 | 0.8871 |
| G temporal inf | 10413 | 0.4588 | 0.8871 |
| G temporal middle | 10636 | 0.2893 | 0.7717 |
| Lat Fis-ant-Horizont | 9850 | 0.9504 | 0.9903 |
| Lat Fis-ant-Vertical | 10770 | 0.2109 | 0.6503 |
| Lat Fis-post | 9394 | 0.4754 | 0.8871 |
| Pole occipital | 11616 | 0.0139 | 0.3582 |
| Pole temporal | 8950 | 0.1776 | 0.6503 |
| S calcarine | 10155 | 0.7097 | 0.8942 |
| S central | 9336 | 0.4257 | 0.8871 |
| S cingul-Marginalis | 10181 | 0.6823 | 0.8871 |
| S circular insula ant | 9885 | 0.9903 | 0.9903 |
| S circular insula inf | 9433 | 0.5105 | 0.8871 |
| S circular insula sup | 9844 | 0.9436 | 0.9903 |
| S collat transv ant | 8869 | 0.1432 | 0.6503 |
| S collat transv post | 10180 | 0.6833 | 0.8871 |
| S front inf | 10947 | 0.1326 | 0.6503 |
| S front middle | 10044 | 0.8308 | 0.9459 |
| S front sup | 9804 | 0.8982 | 0.9775 |

(*Continued*)

**Table 2.** (Continued)

| Region label | Mann-Whitney | | |
| --- | --- | --- | --- |
| | W | Uncorrected p | FDR-corrected p |
| S interm prim-Jensen | 8996 | 0.1997 | 0.6503 |
| S intraparietal and P trans | 11371 | 0.0349 | 0.5161 |
| S oc middle and Lunates | 10247 | 0.5876 | 0.8871 |
| S oc sup and transversal | 10607 | 0.3086 | 0.7717 |
| S occipital ant | 8928 | 0.1677 | 0.6503 |
| S oc-temp lat | 9210 | 0.3287 | 0.7717 |
| S oc-temp med and Lingual | 8557 | 0.0562 | 0.6503 |
| S orbital lateral | 9340 | 0.4290 | 0.8871 |
| S orbital med-olfact | 9551 | 0.6246 | 0.8871 |
| S orbital-H Shaped | 9756 | 0.8442 | 0.9466 |
| S parieto occipital | 10339 | 0.5253 | 0.8871 |
| S pericallosal | 10202 | 0.6604 | 0.8871 |
| S postcentral | 10923 | 0.1417 | 0.6503 |
| S precentral-inf-part | 9606 | 0.6812 | 0.8871 |
| S precentral-sup-part | 9281 | 0.3815 | 0.8554 |
| S suborbital | 9096 | 0.2545 | 0.7532 |
| S subparietal | 10231 | 0.6306 | 0.8871 |
| S temporal inf | 9998 | 0.9232 | 0.9901 |
| S temporal sup | 8257 | 0.0194 | 0.3582 |
| S temporal tranverse | 10908 | 0.1476 | 0.6503 |

period with a paired t-test, smallest uncorrected-p = 0.1771. Additionally, we did not identify any changes in cortical volume in any of the seven networks from baseline to their scan immediately prior to returning to contact, smallest uncorrected-p = 0.1551 (Table 3).

We also compared cortical volume on the region level from baseline to acutely post-concussion and immediately prior to return to contact with paired t-tests. There were no differences for either analysis after multiple comparison correction with FDR (baseline to post-injury smallest FDR-corrected p = 0.632, baseline to return to contact smallest FDR-corrected p = 0.963).

## Discussion

Self-reported concussion history was not associated with reliable differences in baseline symptom reporting or cognitive performance. We found that concussion history was not associated with differences in cortical volume in a large sample of collegiate athletes – either at the level of functional networks or individual regions. Furthermore, we found support for the null hypothesis that concussion history is not associated with cortical volume differences on the network level with Bayesian statistics. These results strongly suggest that there is not a meaningful difference in cortical volume between young adult athletes with and without self-reported concussion history. Despite a lack of difference in cortical volume related to concussion history, we did identify relationships between cortical volume and cognitive performance, with greater cortical volume in the visual network and dorsal attention network being associated with better memory and motor performance. Finally, we examined the possibility that measures of cortical morphology may change in the more acute period following concussion. We did not find any evidence that cortical volume changed on the network or region level between baseline and post-injury (approximately 48 hours following injury), or when participants were cleared to return to contact (largely within 2 weeks of injury).

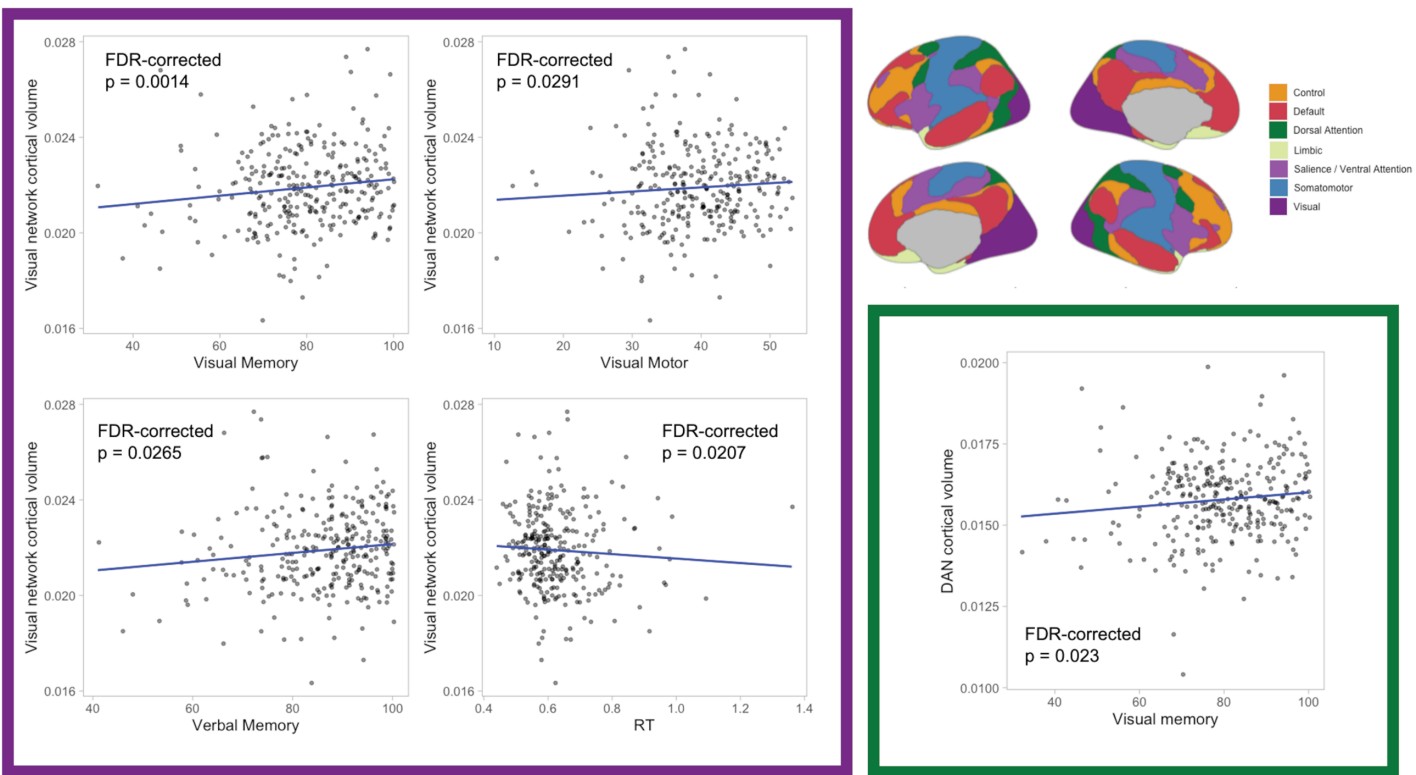

**Fig 4. Cortical volume is related to cognitive performance in the visual and dorsal attention networks.** Visual memory, verbal memory, verbal memory and reaction time composite scores for the ImPACT are correlated with cortical volume in the visual network (purple square). Visual memory performance is correlated with cortical volume in the dorsal attention network (green square).

Previous studies have suggested that concussion history is not related to differences in baseline neurocognitive assessment in college athletes [67–69] and in players participating in the National Football League Scouting Combine [70]. Large meta-analytic studies have provided further support [71], but it should be noted that another meta-analysis did find weak evidence for concussion history being related to worse performance on an attention domain that overlaps with some ImPACT measure scores [72]. We did not identify any differences in the majority of ImPACT composite scores or self-reported symptoms between collegiate athletes with and without concussion history after correcting for multiple comparisons. Reaction time was significantly faster in participants with self-reported concussion history after multiple comparison correction using null hypothesis significance testing. However, Bayesian statistics only suggested anecdotal evidence for this difference. The possible difference in reaction time between those with and without concussion history could be explained by several factors. It is possible that concussion history has a small effect on reaction time. It is also possible that participants with concussion history have completed the ImPACT assessment more frequently (given it is part of clinical care), and the increase in speed for the concussion history group is a practice effect. There is also a possibility that faster reaction time conveys an advantage in sports and that people with faster reaction time get more playing time and are thus at greater risk for concussion. Together, the lack of reliable differences in cognition or symptoms prior concussion history with increased symptoms or cognitive deficits in this sample of collegiate athletes.

**Table 3. Comparison of cortical volume in networks from baseline to acutely post-concussion (48 hrs), and immediately prior to return to contact.**

| Network | Baseline to Post-concussion | | Baseline to Return to Contact | |
|---|---|---|---|---|
| | t | Uncorrected p | t | Uncorrected p |
| Visual | -0.178 | 0.860 | -1.450 | 0.155 |
| Somatomotor | -0.035 | 0.972 | -1.213 | 0.232 |
| Dorsal Attention | 1.372 | 0.177 | 0.889 | 0.379 |
| Ventral Attention | 0.194 | 0.848 | 0.247 | 0.806 |
| Limbic | 0.888 | 0.380 | 0.660 | 0.513 |
| Control | 0.803 | 0.426 | 0.181 | 0.857 |
| Default Mode | -0.428 | 0.671 | -0.573 | 0.570 |

We did not observe any differences in network or region-level cortical volume between participants with and without concussion history. Additionally, we did not observe a difference in network-level cortical volume between individuals reporting zero and those reporting 2 or more previous concussions. These results are consistent with recent, larger sample studies examining the effect of concussion history on cortical morphology on the region- or vertex-level [36,39,40]. Other studies have reported associations between concussion history and morphology, but in many cases, the sample sizes of these studies were much smaller [27–31,73,74], injuries were more severe [26], or participants were experiencing persistent symptoms [75]. We used Bayesian statistics and found moderate evidence for the null hypothesis that concussion history is not associated with differences in cortical volume on the network level. Together, these results add to growing evidence that cortical morphology is not different between those with and without concussion history. Our results are based on a sample of young adult collegiate athletes. It remains possible that the effects of concussion on brain morphology follow a longer temporal trajectory than we were able to measure as suggested by others [76,77], or that intelligence or educational attainment [65] or fitness level can mask more subtle effects [78].

We did find significant relationships between cortical volume in the visual and dorsal attention network and cognitive performance. Specifically, cortical volume in the visual network was positively correlated with visual and verbal memory, and visual motor performance composites from the ImPACT. Cortical volume in the visual network was negatively correlated with the reaction time composite. Finally, cortical volume in the dorsal attention network was positively correlated with visual memory. Other studies have identified relationships between cortical morphology in the visual network and attention [79], general intelligence [80], balance training [81], and visual cognitive task performance [82]. Associations between cortical morphology in the dorsal attention network and cognition have received less attention in the literature, but cortical thickness in the dorsal attention network can be used to predict executive function and memory [83]. Cortical thickness in the dorsal attention network has also been associated with sustained attention [79]. Our results are consistent with this previous work describing relationships between cortical morphology and cognitive performance.

Although we did not observe any differences in cortical volume as a function of concussion history, we measured cortical volume when participants entered their sport at UNL. It is possible that the prior concussions that were reported had occurred years earlier, and that the effect of concussion on brain morphology is rather transient. A subset of our sample (n = 44) was diagnosed with a concussion while at UNL, so we explored effects as a function of acute concussion but did not observe any changes in cortical volume. Previous studies examining the effect of acute concussion on cortical morphology are mixed. Some studies report

increases in cortical volume or thickness in the period immediately following concussion [32, 35], others report a mix of decreases and increases in cortical volume [34], and no changes have also been reported [37]. Some of these inconsistencies may be related to differences in samples (e.g., military, collegiate athletes, community), time points evaluated (e.g., when neuroimaging data was collected relative to injury), and methodological differences (e.g., choice of control group, atlas used). While our acute concussion sample size is modest, our study was unique in its examination of volumetric changes from prior to concussion to acutely after concussion (within roughly 48 hours of injury) in the same participants. This within-subjects assessment decreases the probability that potential structural differences attributed to concussion could be explained by variability between independent concussion and non-concussion control groups. Together, not only is self-reported concussion history not related to cortical volume in collegiate-aged participants, but cortical volume does not appear to change in the more acute period surrounding injury.

## Limitations

The current study has some limitations. First, our sample largely consisted of male football players (only a small subset of our sample comprised female soccer players). Future studies may want to increase sampling of female participants to ensure that results are generalizable, or to accurately describe potential differences in the relationship between concussion history and brain morphology between males and females. Second, our sample of participants with concussion history were not experiencing more symptoms or cognitive deficits at baseline. It is possible that changes in brain morphology may be related to persistent or untreated effects of concussion, and selectively targeting those experiencing persistent symptoms may reveal structural brain changes. Third, our measure of concussion history was collected as part of the Department of Athletics concussion management team's intake process. It was a self-report measure that did not specify if the concussions were related to sport participation, other factors, or when the concussion occurred. While reliability of self-report concussion history is relatively high [84], there is room for inconsistencies which could be random or systematic in nature. Additionally, these self-reported diagnoses were made by different practitioners who may be following different protocols. It is possible that the self-report measure is influenced by diagnoses which may not have been accurate. It is also possible that the mechanism of injury is an important factor to consider and that cannot be factored into the present results. Future studies should adapt well-validated approaches to collecting a detailed concussion history [85].

## Conclusions

Together, our results suggest that concussion history in young adult collegiate athletes is not associated with changes in cortical volume. We found that participants with and without concussion history reported similar (minimal) levels of baseline symptoms and were characterized by similar levels of cognitive performance at baseline. We did not observe differences in cortical volume at the network- or region-level, even when we tested the more extreme comparison of participants reporting two or more concussions to those reporting no concussion history. We did find significant relationships between cortical volume and cognitive performance in the visual and dorsal attention networks, but this was independent of concussion history. Finally, we did not observe any differences in cortical volume from baseline to either acutely following concussion or from baseline to return to contact. While it is challenging to interpret null findings, this pattern of results is consistent with clinical and other scientific evidence suggesting that concussion-related outcomes are associated with disruptions in brain

function and connectivity as opposed to anatomical changes. Indeed, previous research has suggested that concussion history is associated with other changes in the brain (white matter, functional connectivity, electrophysiology, etc.). Our results suggest that future work to identify brain markers of concussion history may be better served by examining these measures rather than focusing on cortical morphology. Future morphological studies of concussion may consider examining longer-term changes by using older populations. Additional studies could also examine the relationship between injury severity and potential morphological changes related to traumatic brain injury.

## Acknowledgments

This collaborative work was made possible by the staff and facilities of the Center for Brain, Biology and Behavior. We'd like to acknowledge the cooperation, collaboration, and contributions of UNL Athletics, which were critical to the project. In addition, we wish to acknowledge the contributions of Joanne Murray, Garrett Schwindt, Liv O'Claire, Ruby Basyouni, and Elliot Carlson related to collecting the MRI data.

## Author contributions

**Conceptualization:** Douglas H. Schultz, Kate L. Higgins, Maital Neta.

**Data curation:** Douglas H. Schultz, Amanda Chiao, Kate L. Higgins.

**Formal analysis:** Douglas H. Schultz, Michelle C. Barbot.

**Funding acquisition:** Cary R. Savage, Maital Neta.

**Investigation:** Douglas H. Schultz, Heather C. Bouchard, Michelle C. Barbot, Amanda Chiao, Kate L. Higgins.

**Methodology:** Douglas H. Schultz, Heather C. Bouchard.

**Project administration:** Douglas H. Schultz, Amanda Chiao, Kate L. Higgins, Cary R. Savage, Maital Neta.

**Supervision:** Douglas H. Schultz, Heather C. Bouchard.

**Validation:** Douglas H. Schultz.

**Visualization:** Douglas H. Schultz.

**Writing – original draft:** Douglas H. Schultz.

**Writing – review & editing:** Douglas H. Schultz, Heather C. Bouchard, Michelle C. Barbot, Julia M. Laing-Young, Amanda Chiao, Kate L. Higgins, Cary R. Savage, Maital Neta.

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
