## [Decision Letter · Decision Letter 0]

13 Dec 2024

PONE-D-24-14875Self-reported concussion history is not related to cortical volume in college athletes.PLOS ONE

Dear Dr. Schultz,

Thank you for submitting your manuscript to PLOS ONE. After careful consideration, we feel that it has merit but does not fully meet PLOS ONE’s publication criteria as it currently stands. Therefore, we invite you to submit a revised version of the manuscript that addresses the points raised during the review process.

We look forward to receiving your revised manuscript.

Kind regards,

Monika Błaszczyszyn

Academic Editor

PLOS ONE

Journal Requirements:

[Funding for this study was provided by the Great Plains IdeA-CTR (MN) and the UNL Office for Research and Economic Development].

3. In the online submission form, you indicated that [Deidentified data from this study are not available in a public archive. Deidentified data from this study will be made available (as allowable according to institutional IRB standards) by emailing the corresponding author].

4. Please update your submission to use the PLOS LaTeX template. The template and more information on our requirements for LaTeX submissions can be found at http://journals.plos.org/plosone/s/latex.

Reviewers' comments:

Reviewer's Responses to Questions

**Comments to the Author**

1. Is the manuscript technically sound, and do the data support the conclusions?

Reviewer #1: Yes

Reviewer #2: Yes

2. Has the statistical analysis been performed appropriately and rigorously? 

Reviewer #1: Yes

Reviewer #2: Yes

3. Have the authors made all data underlying the findings in their manuscript fully available?

Reviewer #1: Yes

Reviewer #2: No

4. Is the manuscript presented in an intelligible fashion and written in standard English?

Reviewer #1: Yes

Reviewer #2: Yes

5. Review Comments to the Author

Reviewer #1: Well written and properly conducted research. This topic is of interest to the area of publication. There are a number of limitations, but they have been properly addressed in the paper. Statistical analysis was appropriate

Reviewer #2: The manuscript presents a thorough investigation of the relationship between concussion history and cortical volume in collegiate athletes. Some positive aspects of the paper were that design is robust, particularly in terms of MRI acquisition and the use of both region- and network-level analyses. The null results are well-supported by the data, and the inclusion of Bayesian statistics strengthens the reliability of these findings. The literature contains a variety of findings on the topic, and a more detailed discussion of potential reasons for the lack of observed differences would improve the manuscript.

While the general topic, studying the relationship between concussion history and cortical volume, is interesting, and make the paper potentially publishable. However, I do have some concerns that I will list below.

Abstract: The abstract would benefit from including essential details such as the age range of the participants and a clear description of the methodology used for data collection. While the number of participants is mentioned, the lack of these additional details leaves the reader without crucial information about the study's design and population. Including this will provide a more comprehensive overview of the study.

Introduction: The introduction requires an additional opening paragraph that contextualizes the general area of research. This paragraph should define a concussion, explain its impact on individuals, and highlight its occurrence in sports-related activities. Establishing this foundational knowledge is critical to emphasizing the importance of studying concussions.

Furthermore, the final paragraph of the introduction should be restructured. It is recommended to split it into two paragraphs for better coherence. Specifically, the argument beginning with "The goal of the current project was to..." would benefit from more precise development, with a conclusion that directly supports the study's significance. In addition, when referencing "Few studies have examined...," it would be more robust to provide specific examples or references to relevant studies, offering readers a clearer understanding of the existing research landscape. Briefly mentioning the study's methodology in this section would enhance its clarity.

Methods: The methods section is well-detailed, but some improvements can be made. For example, in the section that discusses the statistical methods, it would be helpful to specify the tools used for analysis (e.g., IBM SPSS or other relevant software). Additionally, the period over which the data was collected should be clearly stated, as this is important for understanding the study's temporal scope and reliability.

Conclusion: While the study's value is evident, the conclusion could be further strengthened by expanding on future research opportunities. Rather than offering just one suggestion, exploring additional areas for further study would be beneficial, as this will provide more direction for ongoing research in this field.

6. PLOS authors have the option to publish the peer review history of their article (what does this mean?). If published, this will include your full peer review and any attached files.

Reviewer #1: **Yes: **Eric Schussler

Reviewer #2: No

---

## [Author Response · Author response to Decision Letter 1]

16 Jan 2025

We thank the reviewers for their time and attention to the manuscript. We believe the manuscript has improved based on the modifications we made in response to the reviews. In this document the original reviewer comments appear in standard font. Our responses are provided in italics. When appropriate we have included the changes we made to the text by showing them in red in this document.

Regarding the PLOS Data policy: Our data cannot be deposited in a public repository as it would breach compliance with the protocol approved by our research ethics board.

Reviewer #1: Well written and properly conducted research. This topic is of interest to the area of publication. There are a number of limitations, but they have been properly addressed in the paper. Statistical analysis was appropriate.

We appreciate the review and thank the reviewer for their time and attention to the manuscript.

Reviewer #2: The manuscript presents a thorough investigation of the relationship between concussion history and cortical volume in collegiate athletes. Some positive aspects of the paper were that design is robust, particularly in terms of MRI acquisition and the use of both region- and network-level analyses. The null results are well-supported by the data, and the inclusion of Bayesian statistics strengthens the reliability of these findings. The literature contains a variety of findings on the topic, and a more detailed discussion of potential reasons for the lack of observed differences would improve the manuscript.

While the general topic, studying the relationship between concussion history and cortical volume, is interesting, and make the paper potentially publishable. However, I do have some concerns that I will list below.

We appreciate the review and thank the reviewer for their time and attention to the manuscript. We have addressed the concerns below and modified the manuscript accordingly. We believe that these changes have helped improve the manuscript.

Abstract: The abstract would benefit from including essential details such as the age range of the participants and a clear description of the methodology used for data collection. While the number of participants is mentioned, the lack of these additional details leaves the reader without crucial information about the study's design and population. Including this will provide a more comprehensive overview of the study.

We agree that adding details to the abstract would help provide a more comprehensive overview of the study. We added as much detail as possible within the confines of the abstract word limit. The text we added is included below:

“We collected baseline behavioral data including the Immediate Post-Concussion Assessment and Cognitive Testing, a self-report measure of the number of diagnosed concussions, and structural MRI in college athletes (n=296; 263 men and 33 women, age range 17-24). Behavioral measures were collected by members of the Department of Athletics concussion management team using a standardized protocol as part of their on-boarding process.”

Introduction: The introduction requires an additional opening paragraph that contextualizes the general area of research. This paragraph should define a concussion, explain its impact on individuals, and highlight its occurrence in sports-related activities. Establishing this foundational knowledge is critical to emphasizing the importance of studying concussions.

Furthermore, the final paragraph of the introduction should be restructured. It is recommended to split it into two paragraphs for better coherence. Specifically, the argument beginning with "The goal of the current project was to..." would benefit from more precise development, with a conclusion that directly supports the study's significance. In addition, when referencing "Few studies have examined...," it would be more robust to provide specific examples or references to relevant studies, offering readers a clearer understanding of the existing research landscape. Briefly mentioning the study's methodology in this section would enhance its clarity.

We believe that the opening paragraph does contextualize the general area of research. We open with a statement highlighting the frequency of sport-related concussion. We have added a sentence describing common concussion symptoms (see below for the new text). We then transition to defining concussion history, the central topic of the manuscript, before highlighting the possible impact of concussion history on individuals. If the reviewer has more specific suggestions, we would be happy to consider those.

“Concussion can result in cognitive, somatic, affective, and sleep symptoms.”

We do not fully understand the comment regarding the sentence "The goal of the current project was to..." We clearly describe the goal of the current project, “…to determine the effects of self-reported concussion history on brain morphology, specifically examining how network-level changes in cortical volume were related to concussion history in a large sample of collegiate athletes.” Having said that, we did make some changes to this section based on the reviewer’s subsequent comment (see below) that further clarify the significance of the study.

We agree with the reviewer that the sentence referencing, “Few studies have examined...,” is vague. We modified the sentence to focus on what the majority of previous studies have done and in the next sentence we emphasize the significance of our study focusing on the network level. The modifications can be seen below:

“Most previous studies have examined brain morphology in the context of concussion history at the vertex level or in regions of interest.”

Methods: The methods section is well-detailed, but some improvements can be made. For example, in the section that discusses the statistical methods, it would be helpful to specify the tools used for analysis (e.g., IBM SPSS or other relevant software). Additionally, the period over which the data was collected should be clearly stated, as this is important for understanding the study's temporal scope and reliability.

We have added these details to the Methods section:

A statement about the period of time when data was collected: “Data was collected from June 2018 through May 2022.”

A statement about the software used for region-level analyses in the MRI Preprocessing section: “Region-level statistical tests were conducted in Matlab and JASP.”

A statement about the software used for network-level analyses in the MRI Network Analysis section: “Network-level statistical tests were conducted in Matlab and JASP.”

Conclusion: While the study's value is evident, the conclusion could be further strengthened by expanding on future research opportunities. Rather than offering just one suggestion, exploring additional areas for further study would be beneficial, as this will provide more direction for ongoing research in this field.

We agree with the reviewer that touching on additional areas for further study would be beneficial in the Conclusions. These topics were briefly introduced in the Limitations, but we have now expanded on some of those ideas more explicitly in the Conclusions:

“Future morphological studies of concussion may consider examining longer-term changes by using older populations. Additional studies could also examine the relationship between injury severity and potential morphological changes related to traumatic brain injury.”

---

## [Decision Letter · Decision Letter 1]

7 Feb 2025

Self-reported concussion history is not related to cortical volume in college athletes.

PONE-D-24-14875R1

Dear Dr. Schultz,

We’re pleased to inform you that your manuscript has been judged scientifically suitable for publication and will be formally accepted for publication once it meets all outstanding technical requirements.

Kind regards,

Monika Błaszczyszyn

Academic Editor

PLOS ONE

Reviewers' comments:

Reviewer's Responses to Questions

**Comments to the Author**

1. If the authors have adequately addressed your comments raised in a previous round of review and you feel that this manuscript is now acceptable for publication, you may indicate that here to bypass the “Comments to the Author” section, enter your conflict of interest statement in the “Confidential to Editor” section, and submit your "Accept" recommendation.

Reviewer #2: All comments have been addressed

2. Is the manuscript technically sound, and do the data support the conclusions?

Reviewer #2: Yes

3. Has the statistical analysis been performed appropriately and rigorously? 

Reviewer #2: Yes

4. Have the authors made all data underlying the findings in their manuscript fully available?

Reviewer #2: Yes

5. Is the manuscript presented in an intelligible fashion and written in standard English?

Reviewer #2: Yes

6. Review Comments to the Author

Reviewer #2: Thank you for your detailed responses to my comments and for addressing the suggested revisions. I have reviewed your responses and the modifications made to the manuscript, and I find them satisfactory.

I appreciate your effort in improving the quality of the paper, and I approve the revisions. Wishing you success with your research and publication.

7. PLOS authors have the option to publish the peer review history of their article (what does this mean?). If published, this will include your full peer review and any attached files.

Reviewer #2: No

---

## [Editor Report · Acceptance letter]

PONE-D-24-14875R1

PLOS ONE

Dear Dr. Schultz,

I'm pleased to inform you that your manuscript has been deemed suitable for publication in PLOS ONE. Congratulations! Your manuscript is now being handed over to our production team.

Kind regards,

on behalf of

Dr. Monika Błaszczyszyn

Academic Editor

PLOS ONE